

# High gas-phase mixing ratios of formic and acetic acid in the High Arctic

Emma L. Mungall[1], Jonathan P.D. Abbatt[1], Jeremy J.B. Wentzell[2], Gregory R. Wentworth[1,3], Jennifer G. Murphy[1], Daniel Kunkel[4], Ellen Gute[1], David W. Tarasick[2], Sangeeta Sharma[2], Christopher J. Cox[5,6], Taneil  Uttal[5], and John Liggio[2]

[1]Department of Chemistry, University of Toronto, Toronto, Canada
[2]Air Quality Processes Research Section, Environment and Climate Change Canada, Toronto, Ontario, Canada
[3]Environmental Monitoring and Science Division, Alberta Environment and Parks, Edmonton, Alberta, Canada
[4]Atmospheric Physics, Johannes Gutenberg University of Mainz, Mainz, Germany
[5]NOAA Earth Systems Research Laboratory (ESRL), Physical Sciences Division (PSD), Boulder, CO, USA
[6]Cooperative Institute for Research in Environmental Sciences (CIRES), Boulder, CO, USA

*Correspondence to:* Jon Abbatt (jabbatt@chem.utoronto.ca)

**Abstract.** Formic and acetic acid are ubiquitous and abundant in the Earth's atmosphere and are important contributors to cloud water acidity, especially in remote regions. Their global sources are not well understood, as evidenced by the inability of models to reproduce the magnitude of measured mixing ratios, particularly at high northern latitudes. The scarcity of measurements at those latitudes is also a hindrance to understanding these acids and their sources. Here, we present ground-based gas-phase

measurements of formic acid (FA) and acetic acid (AA) in the Canadian Arctic collected at 0.5 Hz with a high resolution chemical ionization time-of-flight mass spectrometer using the iodide reagent ion (Iodide HR-ToF-CIMS, Aerodyne). This study was conducted at Alert, Nunavut, in the early summer of 2016. FA and AA mixing ratios for this period show high temporal variability and occasional excursions to very high values (up to 11 and 40 ppbv respectively). High levels of FA and AA were observed under two very different conditions: under overcast, cold conditions during which physical equilibrium

partitioning should not favour their emission, and during warm and sunny periods. During the latter, sunny periods, the FA and AA mixing ratios also displayed diurnal cycles in keeping with a photochemical source near the ground. These observations highlight the complexity of the sources of FA and AA, and suggest that current chemical transport model implementations of the sources of FA and AA in the Arctic may be incomplete.

## 1   Introduction

Formic acid (FA) and acetic acid (AA) are ubiquitous and abundant in the troposphere and are major contributors to cloud water acidity in remote regions (Paulot et al., 2011). Cloud water serves as an important atmospheric chemical reactor (Lelieveld and Crutzen, 1991). As many chemical reactions depend strongly on pH, cloud water acidity is relevant to the formation of secondary organic aerosol as well as to the processing of other forms of atmospheric particulate matter (Ervens et al., 2011). FA and AA have been measured in a wide variety of environments, both urban and remote, from the poles to the Amazon

to the largest cities in North America. FA and AA have many different sources, although the relative contributions of those

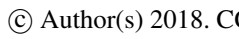



sources depends on the location, and not every source will be relevant in every location. The sources of FA and AA include secondary photochemical production from both anthropogenic and biogenic precursors (Yuan et al., 2015; Liggio et al., 2017); direct emission from plants (Kesselmeier et al., 1998; Kuhn et al., 2002), soils (Sanhueza and Andreae, 1991), and biomass burning (Veres et al., 2010; Ito and Penner, 2004); direct emissions from fossil fuel combustion (Crisp et al., 2014; Kawamura et al., 1985; Liggio et al., 2017); and photochemical production in snowpacks (Dibb and Arsenault, 2002). FA and AA tend to decrease with altitude (Millet et al., 2015; Paulot et al., 2011), and aircraft studies (Jones et al., 2014; Le Breton et al., 2012, e.g.) generally find lower mixing ratios than ground-based studies (Schobesberger et al., 2016; Talbot et al., 1988, e.g.), suggesting that the major sources of these compounds may be at or near the surface.

While the balance between sources remains uncertain both globally and regionally, a recent bottom-up estimate of the global FA and AA budget concluded that photochemical production from biogenic precursors was by far the largest source, with the remaining sources each contributing an order of magnitude less (Paulot et al., 2011). Furthermore, two pieces of evidence indicate that anthropogenic sources, whether primary or secondary, probably do not account for the majority of atmospheric FA and AA. First, radiocarbon analysis of FA has found it to be mostly composed of modern carbon (Glasius et al., 2001), particularly in non-urban areas. Second, analyses of ice cores from the Greenland ice sheet spanning the last 10,000 years have found concentrations of formate and acetate similar to modern snowpack concentrations, with lower values during the last ice age, suggesting the importance of biogenic contributions from North America (Legrand and De Angelis, 1995).

In contrast to their sources, the sinks of FA and AA are thought to be fairly well constrained. The reactivity of these acids towards the hydroxyl radical is low, while their Henry's Law constants are high, such that their main sink in the boundary layer is deposition, both wet and dry (Paulot et al., 2011). Once taken up into water, FA and AA are degraded both photochemically and microbially. FA and AA can also be taken up onto dust and may undergo heterogeneous degradation there. Gas-to-particle partitioning is not considered an important sink of FA and AA as particle-phase concentrations are generally only a few percent of gas-phase mixing ratios (Liu et al., 2012; Chebbi and Carlier, 1996; Baboukas et al., 2000). Considering the various sinks, the lifetimes of FA and AA in the boundary layer have been estimated at 1–2 days (Paulot et al., 2011). Despite all of this information, chemical transport models and box models persistently underestimate the FA and AA mixing ratios measured by both remote sensing and in situ techniques (Paulot et al., 2011; Millet et al., 2015; Schobesberger et al., 2016; von Kuhlmann, 2003; Liggio et al., 2017; Yuan et al., 2015; Stavrakou et al., 2012).

Several explanations for the discrepancy between measurements and models have been suggested, including an unconstrained soil source (Schobesberger et al., 2016), a direct biogenic source which exceeds prior expectations (Stavrakou et al., 2012; Schobesberger et al., 2016; Millet et al., 2015), and as-yet-unknown secondary chemistry (Liggio et al., 2017; Paulot et al., 2011; Millet et al., 2015). These unexplained mixing ratios are often high (on the order of a few ppbv), and the model-measurement discrepancies are particularly large in the Arctic and northern mid-latitudes (Paulot et al., 2011; Schobesberger et al., 2016; Stavrakou et al., 2012). Despite in situ measurements at high latitudes showing significant amounts of FA and AA (high pptv to low ppbv levels) (Talbot et al., 1992; Klemm et al., 1994; Dibb and Arsenault, 2002; Jones et al., 2014; Viatte et al., 2014), the modeled concentrations for both acids in this region are very low. While some of this discrepancy may be due to a failure of the models to capture the persistent stable stratification which characterises the Arctic boundary layer (Tjern-




ström, 2007; Willis et al., 2017), problems with the representation of precursor emissions, chemical production, or emission processes are also likely contributors. Underestimating FA and AA mixing ratios is problematic because of the important role FA and AA are expected to play in determining rainwater acidity in the Arctic and other remote environments.

The existing high-latitude studies are widely separated in time and space. In addition to their sporadic nature, none of these studies report a continuous high-temporal-resolution time series measured at a ground site. Such measurements are valuable for isolating competing atmospheric processes. Here, we present measurements of FA and AA made with an iodide HR-ToF-CIMS over three weeks spanning the transition from late spring to mid-summer (June 18 - July 13, 2016) at Alert, Nunavut, as well as measurements of formate and acetate in precipitation over the same time period. Given the scarcity of data in the region, this study constitutes an important addition to our understanding of the global distribution of these compounds. We discuss the possible sources of the measured FA and AA. Since these acids have frequently been shown to share similar sources and are generally discussed together (Paulot et al., 2011; Baboukas et al., 2000, e.g), and in this work are well correlated ($R^2$ = 0.63), we do not separate them as we attempt to place our observations within the context of the current understanding of their sources.

## 2 Methods

### 2.1 Field site

The sampling campaign took place from 18 June 2016 - 13 July 2016 at the Dr. Neil Trivett Global Atmospheric Watch Observatory (GAW lab) at Alert, Nunavut, located near the northern tip of Ellesmere Island (82°27'N 62°30'W, 187 m elevation). The GAW lab is located 7 km southwest of Canadian Forces Station Alert (relative locations of the station and the laboratory are shown in Fig. 1a). The GAW lab experienced 24 hours of sunlight throughout the campaign.

### 2.2 Gas-phase measurements

FA and AA were measured using a high resolution time-of-flight chemical ionization mass spectrometer (HR-ToF-CIMS, Aerodyne) with the iodide reagent ion. The iodide CIMS ionizes analyte molecules by means of clustering with the $I^-$ ion to form a charged adduct (Lee et al., 2014). The iodide CIMS was located in a wooden shed set up on scaffolding outside the main laboratory building. A 3/8" OD Teflon inlet (length 20') sampled from a height 15' above the ground, where it was secured to the second level of the scaffolding. A bypass flow of 25 slpm, controlled by a critical orifice, passed through this line, from which the CIMS subsampled at the rate of 2.3 slpm (also controlled by a critical orifice). These parameters resulted in a residence time of air in the inlet of ∼1 second. The iodide reagent ion was supplied by a flow of nitrogen from a nitrogen generator (Parker-Balston UHP3200CN2) passing over a permeation tube containing methyl iodide. The permeation tube was contained within a permeation oven and the temperature of the outer surface of the permeation oven was maintained at 40°C (the temperature in the center of the permeation oven would have been lower, but was not measured). The methyl iodide-containing nitrogen stream passed through a sealed $^{210}$Po radioactive source, creating $I^- \cdot H_2O$ ion clusters which reacted with



**Figure 1.** Photographs showing important aspects of the sampling environment. a) Google Earth photograph showing CFS Alert and the GAW lab. b) Photograph from the GAW lab in late June 2017 showing the snowpack remnants. c) Photograph from the GAW lab in early July 2017 showing that all the snow had melted. d) Characteristic close-up of the surrounding tundra showing the large percentage of bare soil.

the molecules of interest in the ion-molecule reaction region (IMR) of the mass spectrometer. Analytes are detected as their $I^-$ clusters. Backgrounds were collected for 2 minutes of every hour by overflowing the mass spectrometer inlet (downstream of the Teflon inlet line) with the exhaust flow from the bypass line, which has been passed over a Pt catalyst heated to 350°C, followed by sodium bicarbonate and activated carbon cartridges in series. The instrumental stability was monitored by constant

5   addition of $^{13}$C labelled propionic acid.

Calibrations were carried out after the campaign using the Ionicon Liquid Calibration Unit (LCU). Known mixing ratios of FA and AA at known absolute humidities were provided to the instrument by the LCU. The humidity dependence of the individual compound sensitivities was determined, as water vapor can cluster preferentially with the iodide ions, decreasing the ionization efficiency of analyte molecules. These calibration data were fit using a multiple linear regression and applied to

10   the ambient data as a function of the absolute humidity measured in the flow exiting the IMR to correct for the water vapor





dependence. Details concerning the calibration calculations may be found in the Supporting Information (Section S1). The limits of detection were, on average, 0.3 ppbv for FA and 0.7 ppbv for AA. The limits of detection were calculated as three times the standard deviation in the background signal averaged over a five minute period, and varied by less than $10\,\%$ over the entire campaign. The calibration uncertainty (determined as the range between the two calibration experiments) was $15\,\%$ for

FA and $50\,\%$ for AA.

### 2.3  Precipitation measurements

Bulk precipitation samples were collected on the roof of the GAW lab using three adjacent collectors consisting of a polypropylene funnel (0.1 m diameter) secured to a polypropylene bottle (125 mL total volume). Bulk collectors were rinsed three times each with deionized water and air dried prior to deployment. We assume the input of dry deposition to be negligible given

the low particulate matter concentration throughout the study as well as the small diameter of the funnel mouth. Precipitation samples were collected at the end of each precipitation event or, if precipitation occurred overnight, at 8:30 (local time) the following morning. Samples were immediately frozen to prevent biodegradation (Vet et al., 2014) until analysis at the end of the campaign. Acetate and formate were quantified using ion chromatography (IC, Thermo Fisher Scientific) that was calibrated with a series of 5 aqueous standards prepared via serial dilution. A field blank was collected by rinsing each collector with

$\sim 50\,\mathrm{mL}$ of deionized water. The volume-weighted IC signal of the field blank was subtracted from each sample. However, the field blank signals for formate and acetate were roughly a factor of 50 and 100 smaller, respectively, than the average precipitation sample. The pH of the samples was not measured. The pH of the precipitation was estimated using IC measurements of the major ions ($\mathrm{Na^+}$, $\mathrm{NH_4^+}$, $\mathrm{Cl^-}$, $\mathrm{NO_3^-}$, $\mathrm{SO_4^{2-}}$) as $pH = -log\left(\sum anions - \sum cations\right)$.

The duration of the sampled precipitation events was estimated using the images captured every minute by an upward

facing camera (Total Sky Imager TSI-440, Yankee Environmental Systems, Inc.). Precipitation is visible in these images as raindrops striking the lens (e.g. Fig. S7), and the time that the rain stopped was identified as when the pattern of drops stopped changing. Precipitation amount was quantified by dividing the volume of each precipitation sample by the funnel footprint of the collectors (details in Supporting Information, Section S3.1).

### 2.4  Ancillary data

Aerosol particles were brought into the GAW lab from the ambient atmosphere via a 3 m long 10 cm inner diameter vertical manifold with a $\sim 1000$ lpm flow rate. The particles were subsampled for individual instruments by stainless steel tubing from the center of the flowstream, about 30 cm up from the bottom of the manifold. The mean residence time of a particle in the sampling manifold before being detected by an instrument was 3 s. The total particle counts greater than 4 nm were measured by a Condensational Particle Counter (CPC, TSI 3772). Particle diameters from 20 nm to 500 nm were measured

using a Scanning Mobility Particle Sizer (SMPS, TSI 3034). The SMPS was verified using monodisperse polystyrene latex and ammonium sulfate particles using a Scanning Electrical Mobility Spectrometer (SEMS, Brechtel Manufacturing Incorporated).

Wind speed and direction were measured by a wind sensor (Campbell Scientific, 05103-10), and dew point and temperature were measured by a Vaisala logger. All meteorological data are averaged to 5 minutes before saving by the Campbell Scientific




data logger. Total downwelling shortwave radiation (SWD) was measured at a Baseline Surface Radiation Network (BSRN) (Ohmura et al., 1998) station, located approximately 100 m south of the GAW lab. In this work, SWD is represented as the sum of the measured diffuse and direct components as observed at a height of 2 m above the surface. The diffuse was measured by a shaded Eppley Black & White pyranometer (PSP) and the direct was measured by an Eppley normal incidence pyrheliometer (NIP). The data were quality controlled following the method of (Long and Shi, 2008). The calibration uncertainty is 0.5-1% for the NIP and 1-2% for the PSP. Fractional sky cover was derived from the radiation measurements following (Long et al., 2006). Mixing height was estimated from radiosondes deployed by the EC weather station at Alert.

### 2.5 Soil data

Soil temperature was monitored by two iButton temperature loggers (Maxim Integrated) placed at a depth of $10\,\mathrm{cm}$ within $25\,\mathrm{m}$ of the GAW lab. Soil cores were collected in triplicate from three separate sites in the surrounding area (up to 5 km away, one on July 8 2016 and the other two on July 15 2016) and analyzed according to (Wentworth et al., 2014). After clearing any surface debris (i.e., vegetation, rocks, pebbles), $10\,\mathrm{cm}$ deep cores were retrieved using a PVC tube ($5.1\,\mathrm{cm}$ inner diameter) and hammer. Cores remained frozen until they were analyzed about one month later. Soil pH was measured in duplicate for each extract using a standard pH electrode (Orion Model 250A, Thermo Scientific) immersed in a 1:1 slurry of soil and deionized water. The pH electrode was triple-point calibrated with commercially available pH standards (4.01, 7.00, and 10.00).

### 2.6 FLEXPART-ECMWF

Air mass histories were computed using the Lagrangian Flexible Particle Dispersion (FLEXPART) model version 10.0 Stohl et al. (2005), which was driven by meteorological analysis fields from the European Centre for Medium-Range Weather Forecasts (ECMWF). For this study, non-interacting particles are released for 24 h and traced back for 10 days. Results were output every 6 h at six different altitude levels with upper level boundaries of 200 m, 500 m, 1000 m, 2000 m, 5000 m, and 10 km. For use in this work, the potential emission sensitivities (PES), representing the sensitivity of the airmass to emissions from the surface at a given location, were averaged over either the entire column (all six altitude levels) or the first $200\,\mathrm{m}$ (first altitude level) and integrated back in time ten days. PES plots shown in this work were further averaged over several days (up to the length of the entire campaign). Maps were generated using the Basemap toolkit for Python 3.6.

## 3 Results & Discussion

### 3.1 General transport pattern affecting the sampling site

FLEXPART-ECMWF results indicate that the air arriving at Alert had spent at least the last ten days within the Arctic circle (Fig. 2a and b), consistent with the established notion of a transport barrier into the Arctic, often referred to as the polar dome (Stohl, 2006; Klonecki et al., 2003). Furthermore, enhanced wet deposition in the Arctic boundary layer in summer tends to reduce the atmospheric lifetime of scavengeable species such as FA and AA, making long-range transport less likely



(Croft et al., 2016). Indeed, aerosol concentrations reach very low levels in the summer Arctic atmosphere as a result of these scavenging processes (Barrie and Barrie, 1990; Croft et al., 2016; Li et al., 1993). Given these characteristics of transport from lower latitudes to Alert, we do not believe that sources such as biomass burning were important contributors to the mixing ratios observed during this campaign.

The complex topography and coastal situation of Alert result in the local meteorology being significantly influenced by mesoscale circulation processes. This influence is apparent in Fig. 2c, which shows a clear division between strong south-westerlies and weak northeasterlies. These wind regimes are characteristic of Alert (Persson and Stone, 2007) and represent processes occurring on the 10-200 km mesoscale. While the high wind speed southwesterlies provide a larger potential for transport than do the low wind speeds of the northeasterlies, from the perspective of potential sources of FA and AA, the
landscape surrounding Alert is fairly homogeneous on that length scale. This is borne out by FLEXPART-ECMWF PES plots (Fig. S4) showing that the source regions during the two wind regimes similarly encompass sea ice, parts of Ellesmere Island, and some open water to the south.

The mixing ratios of FA and AA shown in Fig. 3 (5 minute averages) display two striking features: high variability on two temporal scales (minutes and days) with occasional excursions to very high values (Figs. 3 and 4); and a clear dependence
on wind direction. In Fig. 3, several time periods have been highlighted with an orange bar. These are times of southwesterly flow, which was always associated with high wind speeds, usually displayed low relative humidities, and often coincided with an increase in temperature. During the rest of the campaign the site experienced much lower wind speeds and variable wind directions. These two distinct regimes are visible in the wind rose shown in Fig. 2c.

The observed dependence of the FA and AA mixing ratios on wind direction could potentially reflect regional heterogeneity
in sources, sinks, or in source strength. However, given the consistency in source regions between the two wind regimes and the general regional Arctic nature of the sampled air masses, a focus on local processes in appropriate for this data set. While the specific time variations in FA and AA mixing ratios we observed are inextricably linked to the complex local topography and meteorology, the inferences drawn about possible sources are generalizable to similar High Arctic environments.

### 3.2    Role of meteorology in determining FA and AA mixing ratios

In Figure 3, lower mixing ratios are seen to coincide with the high wind speed regime. Average mixing ratios of FA and AA for the whole campaign as well as averages over both regimes are summarized in Table 1. The mean mixing ratios for the two regimes are statistically different from each other at the 95 % confidence level. The average mixing ratios during the high wind speed regime (0.76 ppbv and 0.33 ppbv for FA and AA, respectively) were a factor of two and four respectively lower than those during the low wind speed regime (1.48 ppbv and 1.46 ppbv for FA and AA, respectively). However, the average
mixing heights as determined from radiosonde measurements were about $450\,\mathrm{m}$ for the high wind speed regime and $200\,\mathrm{m}$ for the low wind speed regime (Supporting Information Section S2). Assuming that FA and AA (or their precursors) are directly emitted, and well-mixed within the boundary layer, the factor of two difference in mixing heights can account for a factor of two difference in mean mixing ratios between these two regimes, suggesting that relative magnitude of sources and sinks may have been similar between regimes for FA. The additional factor of two difference in AA between the two wind regimes





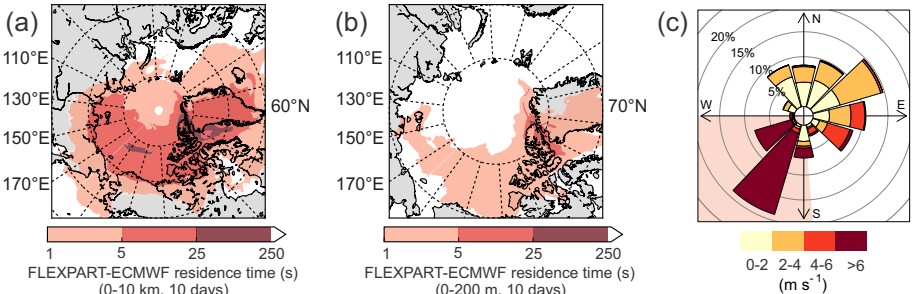

**Figure 2.** a) FLEXPART-ECWMF summed over the entire column (0 km to 10 km), averaged over the entire campaign, and integrated ten days back in time. That the darker colours are mostly restricted to the Arctic indicates that the air arriving at Alert had mostly spent at least the last ten days within the Arctic circle. b) FLEXPART-ECMWF summed over the lowest 200 m, averaged over the entire campaign, and integrated ten days back in time. c) Wind rose for entire campaign. Shaded area shows the high wind speed regime (see text).

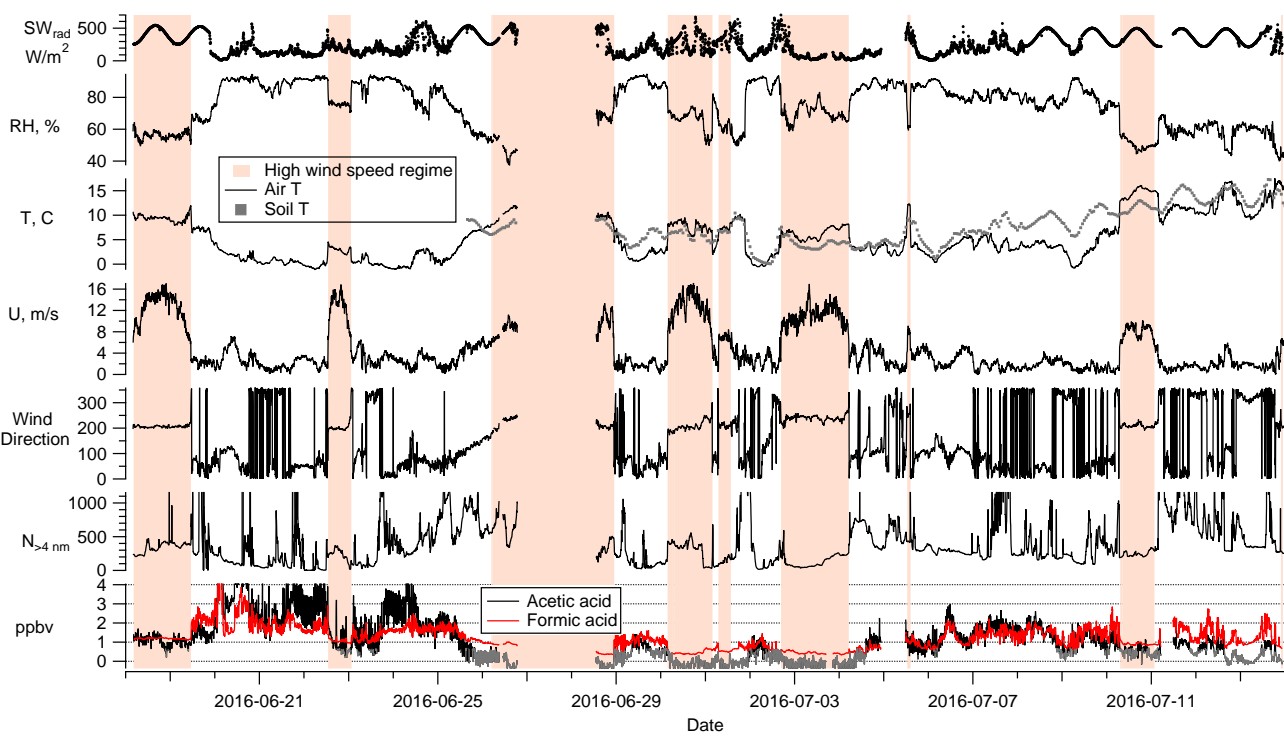

**Figure 3.** Time series of all data collected for the campaign (5 minute averages). From top to bottom: total downwelling shortwave radiation, relative humidity, air temperature and soil temperature, wind speed, wind direction, aerosol particles, and FA (red) and AA (black, with values below the detection limit shown in gray.) The vertical axis is truncated at 4 ppbv for the organic acids and 1200 $\mathrm{cm^{-3}}$ for the aerosol particles. Orange bars highlight the high wind speed regime.





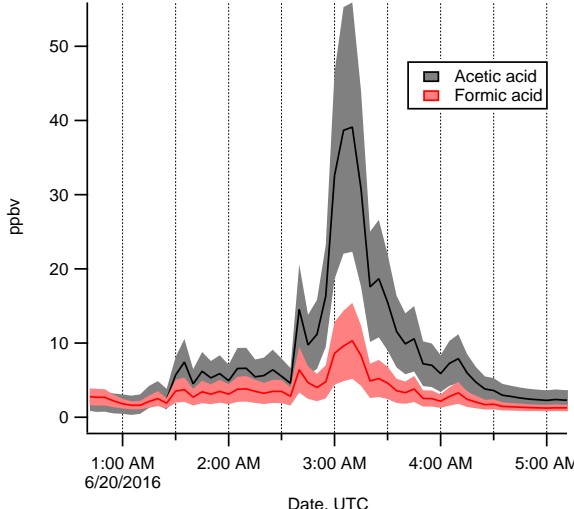

**Figure 4.** Focus on the time of greatest formic and acetic acid measured mixing ratios. Shaded areas indicate calibration uncertainty (range between highest and lowest estimate, Supporting Information Section S1).

is likely spurious. There were fewer high wind events during the latter half of the campaign, when AA was also lower. The decrease in AA across the campaign may indicate a shift to sources with a higher FA to AA ratio. There are some indications that a higher FA to AA ratio is associated with biogenic sources of these acids (Khare et al., 1999; Talbot et al., 1988), but as no conclusive evidence exists we have not attempted to use this changing ratio to further our analysis.

The standard deviations of FA and AA for the entire campaign, the high wind speed regime, and the low wind speed regime are also shown in Table 1. The relative stability of the mixing ratios during the high wind speed regime (as indicated by the smaller standard deviation values) likely reflects the stronger winds promoting mixing and leading to more constant mixing ratios. Thus the most striking behaviour of the mixing ratios (i.e., the difference between the high and low wind speed regimes) is most likely due simply to mixing considerations and is not chemistry or source related. We suggest that the high variability at

short time scales during the low wind speed regime is reflecting sources of FA and AA which are in close enough proximity to the measurement site that the variability has not been smoothed out by mixing processes. The variability also suggests that the sources are likely not only local but heterogeneous and/or sporadic, such that measured mixing ratios will be strongly affected by slight changes in wind direction or fluctuations in source strength.

It is worth noting that compared to mid-summer mixing heights at mid-latitudes, which are typically ∼1 km, the measured

mixing heights for Alert are found to be very low. The low mixing heights are relevant when considering the high magnitudes of the mixing ratios reported here, as dilution is decreased with lower mixing heights. Similar emissions at mid-latitudes might result in a factor of 3 to 4 lower mixing ratios due to dilution into a deeper boundary layer. The high mixing ratios combined with a fairly stable atmosphere are also suggestive of strong vertical gradients of the acid concentrations such as would result from surface sources. Possible candidates for these surface sources are discussed in Section 3.4.



**Table 1.** Mean mixing ratios (in ppbv) of FA and AA for the entire campaign, the high wind speed regime, and the low wind speed regime ± one standard deviation.

|  | Campaign Average | High Wind Speed Regime | Low Wind Speed Regime |
|---|---|---|---|
| FA | 1.23 ± 0.63 | 0.76 ± 0.32 | 1.48 ± 0.64 |
| AA | 1.13 ± 1.54 | 0.33 ± 0.57 | 1.46 ± 1.77 |

## 3.3 Precipitation data

The precipitation data collected during the campaign are summarized in Table 2. Also listed are the calculated depositional fluxes of formate and acetate based on these measurements (Supporting Information Section S3.2). The volume-weighted average concentrations over the entire campaign were $4.3\,\mu$M for AA and $4.2\,\mu$M for FA. Decades of measurements of formate and acetate in rainwater have found the values to range between 0.1 and 33 $\mu$M, with no clear dependence on the type of environment (i.e., urban, rural, remote) (Khare et al., 1999; Talbot et al., 1988, 1992; Kawamura et al., 1996, 2001). The values measured in this study fall within that range and are consistent with studies in remote subpolar areas reporting annual volume-weighted mean concentration ranges from ∼1-8 $\mu$M (see Vet al. (Vet et al., 2014) and references therein).

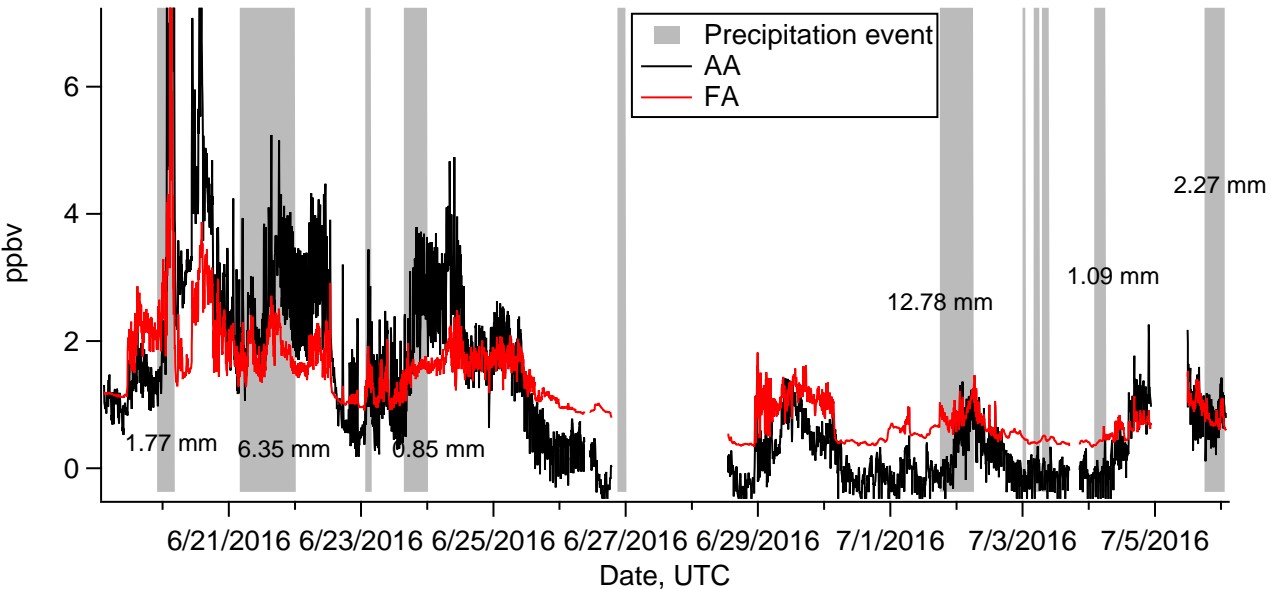

**Figure 5.** Time series FA (red) and AA (black). Gray bars indicate the duration of precipitation events as determined from All Sky Camera images. Precipitation amounts (in mm) are shown where data are available.

The rate of scavenging of highly-soluble gas-phase compounds by precipitation is expected to be on the order of 1-3% per minute (Seinfeld and Pandis, 1998). The estimated durations of the precipitation events which occurred during the campaign





are shown as gray bars in Fig. 5. The precipitation events lasted for at least an hour and usually longer, up to an entire day. FA and AA have very high effective Henry's Law constants at the pH of the precipitation (estimated in Table 2 as $-log[\sum(\text{anions})-\sum(\text{cations})]$). We might thus expect significant depletion of the gas-phase mixing ratios of FA and AA through precipitation scavenging, particularly as the precipitation events were generally associated with low wind speeds, and thus presumably with

little advection of FA and AA (Fig. 3). This expectation can be evaluated by comparing the deposition that would be associated with complete scavenging of the column to the measured deposition (Table 2). Using representative values for the mixing ratio (1 ppbv assuming that the mixing-ratio measured at ground-level is representative of the entire column) and the boundary layer height (200 m, see Section 3.2), we find that the estimated deposition is very similar to the measured deposition ($\sim$ $6\,\mu\text{mol}\,\text{m}^{-2}$, details in Supporting Information Section S3.2). While subject to a number of assumptions and uncertainties,

these calculations nonetheless suggest that at least some scavenging of the column is likely, and should result in decreases of the measured mixing ratios at the ground. However, it is clear from Fig. 5 that this was not observed. Indeed, in many cases the mixing ratios of FA and AA actually increase during the precipitation events, indicating that the FA and AA mixing ratios are being maintained against a depletion process via wet deposition. This observation is suggestive of a surface source of FA and AA to the boundary layer, which may be enhanced during precipitation events. Large enhancements of FA and AA mixing

ratios during or immediately following precipitation events have been reported previously (Yuan et al., 2015; Sanhueza and Andreae, 1991; Warneke et al., 1999; Greenberg et al., 2012; Talbot et al., 1988). Both biological (Sanhueza and Andreae, 1991) and chemical (Warneke et al., 1999; Greenberg et al., 2012) mechanisms have been proposed for enhanced release of FA and/or AA from soil surfaces as a result of precipitation. We will discuss these mechanisms further in Section 3.4.5.

**Table 2.** Precipitation data for FA and AA.

| Sampling Date | June 20 | June 22 | June 24 | June 27 | July 2 | July 4 | July 6 |
|---|---|---|---|---|---|---|---|
| Precipitation type | rain | snow | rain | rain | snow | rain & snow | rain & snow |
| Precipitation (mm) | 1.77 | 6.35 | 0.85 | 1.38 | 12.78 | 1.09 | 2.27 |
| FA concentration ($\mu$M) | 3.2 | 2.3 | 4.7 | 5.6 | 1.9 | 40.0 | 5.9 |
| AA concentration ($\mu$M) | 3.1 | 2.1 | 12.2 | 4.9 | 1.9 | 33.4 | 6.9 |
| FA deposition ($\mu$mol m$^{-2}$) | 5.6 | 14.3 | 4.0 | 7.7 | 23.8 | 43.5 | 13.4 |
| AA deposition ($\mu$mol m$^{-2}$) | 5.5 | 13.5 | 10.4 | 6.7 | 24.5 | 36.3 | 15.7 |
| Estimated pH | 5.0 | 5.5 | 5.7 | 5.7 | 5.3 | 4.2 | 5.3 |





**Table 3.** $R^2$ values for the correlations of the diurnal cycles of FA and AA with solar radiation, wind speed, or both during the period July 5 - July 13.

|  | Solar radiation | Wind speed | Solar radiation + wind speed |
|---|---|---|---|
| FA | 0.29 | 0.01 | 0.80 |
| AA | 0.27 | 0.00 | 0.69 |

## 3.4 Possible sources of FA and AA

Summer is short in the High Arctic, and during the course of this campaign the environment underwent a dramatic shift from late spring (snowpack remaining, little vegetation) to midsummer (very little snow remaining, comparatively extensive vegetation). As the environmental conditions changed, it is likely that the relevant sources of FA and AA also underwent

changes. For this reason, the analysis below frequently divides the campaign in two and discusses the earlier and later portions separately. Time itself is thus an important variable, meaning that it may not be desirable to compare, for example, a sunny day at the beginning of the measurement time period to a sunny day towards the end.

### 3.4.1 Diurnal variability

The amplitude of the diurnal cycles of FA and AA averaged over the entire campaign is very small. The fractional sky cover

differed between the beginning and end of the campaign, with values of 0.69 and 0.36, respectively, during the earlier (June 18 2016 - July 4 2016) and later (July 5 2016 - July 13 2016) portions of the campaign. Restricting our analysis to the less overcast, later period reveals a pronounced diurnal cycle in the acids, in contrast to the first half of the campaign (Fig. 6). However, the diurnal cycles of FA and AA peak a few hours before solar radiation or temperature. This can be accounted for by considering the effects of wind speed. Scaling the diurnal cycles of FA and AA by the diurnal cycle of wind speed produces

a diurnal profile which matches that of solar radiation very well (Fig. 6). Linear regression allows us to quantify how well wind speed and solar radiation can explain the diurnal cycles of FA and AA. Table 3 summarizes the $R^2$ values for the relationships between the acids and solar radiation or wind speed alone or in combination (i.e., for a multiple linear regression of the form $y = m_1 x_1 + m_2 x_2 + b$). These values show that wind speed alone has essentially no explanatory power, solar radiation alone can explain $\sim$30% of the diurnal variation in FA and AA, and solar radiation and wind speed combined can explain $\sim$70-80%

of the diurnal variation in these acids. The explanatory power of solar radiation suggests the presence of a photo-oxidative source during the latter half of the campaign. The modulation of the diurnal cycles of FA and AA by wind speed may indicate a surface source, because a source at the surface would result in an upward gradient of FA and AA. An increase in wind would act to reduce that vertical gradient by mixing the compounds upward, decreasing the mixing ratio measured near the surface. There are two possibilities as to the identity of the photo-oxidative source: heterogeneous oxidation at surfaces or gas-phase

oxidation of reduced compounds, for example of BVOCs. These possible sources will be further discussed in Section 3.4.







**Figure 6.** a) Diurnal cycles of FA, AA, total downwelling shortwave radiation, temperature, and wind speed averaged over the entire campaign. b) Averaged over the period July 5 - July 13. c) The diurnal cycle of FA weighted by that of wind speed (red, left axis) and the diurnal cycle of total downwelling shortwave radiation (black, right axis). d) The relationship between the quantities plotted in (c). Error bars in plots (a) and (b) represent one standard deviation.

### 3.4.2 Anthropogenic emissions

Wind directions during the low wind speed regime were often from the northeast, such that the measurement site was downwind of the military station (CFS Alert) located 7 km to the northeast. These conditions make it important to determine if there is a local anthropogenic source contributing to the FA and AA mixing ratios. Although anthropogenic emissions are not generally

5    thought to be important sources of FA and AA on a global scale, these acids and their precursors certainly have anthropogenic sources in some locales. Activities at CFS Alert that could potentially affect the measured mixing ratios of FA and AA include




the diesel generator powering the station, cooking fumes, and vehicular emissions (although there are less than a dozen vehicles on station). FA and AA are emitted directly from these sources (Crisp et al., 2014; Bannan et al., 2014; Liggio et al., 2017) as well as formed photochemically from these emissions (Liggio et al., 2017; Yuan et al., 2015).

Unfortunately, no chemical tracers that would allow for conclusive evaluation of anthropogenic influence were measured during the campaign. However, each of the combustion sources mentioned above also emits aerosol particles, which *were* measured. We follow two lines of reasoning involving particle concentrations to conclude that combustion sources were not major contributors to the FA and AA measured during this campaign. First, if we consider aerosol number concentrations as a tracer for combustion sources, the lack of correlation ($R^2 < 0.01$) between the number of aerosol particles measured by the CPC (particles >4 nm) and FA and AA argues against a combustion source for the acids. Second, we can use the particle concentrations to evaluate the extent of the dilution a pollution plume originating at the station would have undergone before arriving at the GAW lab. The highest particle concentrations observed at the GAW lab during the campaign were about $10^3$ particles cm$^{-3}$. Combustion sources emit many very small particles which quickly coagulate to yield accumulation mode particle densities on the order of $10^6$ particles cm$^{-3}$ in near-source ambient air (Tang et al., 2016). We conclude that the aerosol particles have been diluted by at least 2 or 3 orders of magnitude before arriving at the GAW lab. The FA and AA mixing ratios observed during the low wind speed regime ($\sim 2$ ppbv) are on the order of those observed in highly polluted urban areas (e.g. the Po Valley, LA; Atlanta (Yuan et al., 2015; Millet et al., 2015)) and close to sources (Bannan et al., 2014). Even accounting for the lower boundary layer at Alert, which might lead to a factor of 5 enhancement in mixing ratios for the same magnitude of emissions, the levels of FA and AA we observed are not consistent with combustion emissions that have been diluted by three orders of magnitude. It is also very unlikely that the aerosol particles could undergo three orders of magnitude of dilution while the acids remained undiluted. Hence the maximum amount of FA or AA that would arrive at the laboratory to be sampled would be on the order of 10 pptv (considering a three order of magnitude dilution of 10 ppbv and accounting for an enhancement in the mixing ratio due to the low mixing height), constituting a minor contribution even to the lower mixing ratios measured during this campaign ($\sim 200$ pptv). We conclude that CFS Alert was not a major contributor to the FA and AA mixing ratios observed during the campaign.

### 3.4.3 Heterogeneous chemistry

Heterogeneous oxidation reactions produce both FA and AA (Vlasenko et al., 2008; Molina et al., 2004), although the mechanism by which this occurs is not yet understood. Of particular relevance to the present study are recent observations of mixing ratios of FA of similar magnitude to those reported here above open water in Nares Strait, only $\sim 300\,\mathrm{km}$ from Alert, which were attributed to heterogeneous chemistry at the sea surface microlayer (Mungall et al., 2017). Significant transport of air masses from the European Arctic and Baffin Bay to Alert does occur (Sharma et al., 2012), raising the possibility that a marine source contributed to the measured mixing ratios of FA and AA. However, there is little open ocean in the vicinity of Alert in June and July 2016 (Fig. S3), and, as will be discussed in the following sections, there were several likely sources of FA and AA much nearer to the point of measurement. Additionally, the environment surrounding the laboratory offered an abundance of other aqueous surfaces. During the snow melt period, shallow ponds form over large areas of the tundra, and the lakes



lose their ice coverage. These natural waters contain photochemically active dissolved organic matter (Laurion and Mladenov, 2013), and as such are good candidates for heterogeneous production of volatile organic compounds (Brüggemann et al., 2017; Ciuraru et al., 2015; Fu et al., 2015). Indeed, the presence of surfactants in these water bodies is indicated by observations of foam at their margins (Fig. S6). Heterogeneous chemistry could also act to oxidize organic molecules present at soil surfaces.

5      Considering the diurnal variation of FA and AA (Section 3.4.1), which was also observed for FA in Nares Strait (Mungall et al., 2017), and the apparent ubiquity of the heterogeneous production of FA and AA, it seems likely that heterogeneous production from soil or water surfaces was a source of FA and AA throughout the campaign. Unfortunately, without further information, it is impossible to assess the extent of the contribution of heterogeneous oxidation processes to the observed FA and AA, although we believe that this source deserves further consideration in future work.

### 10 3.4.4 Snowpack emissions

The campaign took place during the snow melt period. At the beginning of the measurement period, significant snowpack remained (Fig. 1b), but by the end of the measurement period, it had all melted (Fig. 1c). The first half of the campaign, when significant snowpack remained, was also a time of elevated FA and AA mixing ratios (June 20 - June 26, Fig. 3). Arctic snowpacks have been shown previously to give rise to significant mixing ratios of FA and AA (Dibb and Arsenault, 2002) 15 as well as other oxygenated volatile organic compounds (Grannas et al., 2007). The purported mechanism for this release is photochemistry. As the first half of the measurement period was quite dark due to heavy cloud cover (Fig. 3), particularly during times of elevated FA and AA, photochemistry seems unlikely to account for our observations.

Recent work quantifying formate and acetate in fresh snow at Alert found levels of 10 and 20 $\mu$g/kg, respectively (Macdonald et al., 2017). Here we examine the feasibility of the partitioning of FA and AA from melting snow giving rise to the observed 20 mixing ratios. We can use the effective Henry's law constants ($K_H^*$) of FA and AA to calculate their expected partitioning behaviour from the melted snow from Eq. (3)

$$K_H^* = K_H \left( 1 + \frac{Ka}{[H^+]} \right) \tag{1}$$

$$K_H^* = \frac{C_l}{C_g} \tag{2}$$

$$C_g = \frac{C_l}{K_H^*} \tag{3}$$

25  where $C_g$ is the gas phase partial pressure in bar, $C_l$ is the aqueous phase concentration in M, $K_H$ is the Henry's Law constant for a given temperature (Johnson et al., 1996), and Ka is the acid dissociation constant ($1.8 \times 10^{-4}$ for FA and $1.74 \times 10^{-5}$ for AA). Assuming that the concentrations for formate (acetate) and pH measured in freshly fallen snow samples apply to the aged, melting, late-spring snowpack, that melt water would be in equilibrium with approximately 0.1 pptv (3 pptv) in the gas phase, or about 0.001% (0.01%) of the peak measured mixing ratios as are summarized in Table 4.

30      These values are far too small to account for our observations. However, the formate and acetate concentrations used in these calculations are for fresh snow. The aged snowpack present during our measurements might be expected to contain much higher solute concentrations. Additionally, bulk concentrations measured in melted snow likely do not accurately represent the



**Table 4.** Estimated equilibrium mixing ratios for literature values of formate and acetate in snow and soil. Measured values for comparison are higher for the snow case as mixing ratios were higher earlier in the campaign. pH values for snow are for melted fresh snow at Alert (Macdonald et al., 2017). pH values for soil were measured from soil cores in the vicinity of Alert (Section 2.5).

|  | $C_l$ (mg/L) | pH | T | $K_H{}^*$ M hPa$^{-1}$ | Estimated $C_g$, ppbv | Measured $C_g$, ppbv | Percent explained |
|---|---|---|---|---|---|---|---|
| FA (snow) | 0.01 | 5.37 | 0 | $2.4 \times 10^6$ | 0.0001 | 3 | 0.005 |
| AA (snow) | 0.02 | 5.37 | 0 | $1.4 \times 10^5$ | 0.003 | 4 | 0.1 |
| FA (soil) | 2 | 6.5 | 15 | $1.0 \times 10^7$ | 0.004 | 1.5 | 0.3 |
| AA (soil) | 1 | 6.5 | 15 | $4.6 \times 10^5$ | 0.05 | 1.5 | 3.1 |

concentrations present at the surface of the snow in the environment. Concentration might occur in a liquid layer at the edges of snow crystals, or as a result of freeze-thaw cycles; additionally, the first flush of melt water or rain percolating through the snowpack would contain very high concentrations of solutes (Kuhn, 2001; Meyer et al., 2009). The snowpack surrounding the laboratory also experienced frequent precipitation during the time period of elevated mixing ratios (Fig. 5), which would

contribute to snow melt. If these processes led to a decrease in pH below the pKa of FA or AA, the acids would be considerably more likely to partition to the gas phase. However, a three order of magnitude increase in concentration as a result of freeze-thaw cycles or solute exclusion would be required for equilibrium partitioning to explain our observations. Finally, we note that from a mass balance perspective, the snowpack contained sufficient formate and acetate to give rise to the observed high mixing ratios in the extreme case of all of the formate and acetate being released to the gas phase. For example, the complete

evaporation of a 30 cm-deep snowpack containing the formate and acetate values given in Table 4 would yield FA and AA mixing ratios of 8 and 12 ppbv, respectively, if mixed up to a 200 m mixing height.

### 3.4.5 Soil emissions

Direct emissions of FA and AA from soil are highly unconstrained. While FA and AA are known to be produced by soil bacteria (Paulot et al., 2011), a single direct observation of their emission from a soil surface exists (Sanhueza and Andreae, 1991). In

1988, a study in the southeast United States saw that atmospheric FA mixing ratios began to rise in spring before deciduous trees got their leaves and suggested that a soil source might be responsible for these observations, and in 1992 upward gradients of FA and AA from a boreal forest soil were interpreted as indicating a soil source (Enders et al., 1992). More recently, a flux measurement study of FA in the boreal forest concluded that a large direct soil or plant source was needed to explain those measurements (Schobesberger et al., 2016), and a study of the Fenno-Scandinavian Arctic wetlands suggested that that the

acidic, microbially active wetland soils might be a large source of FA (Jones et al., 2017). Given the large expanse of sparsely vegetated ground with a large percentage of bare soil surrounding our measurement site (e.g. Fig. 1d), we will consider the ability of soil emissions to contribute to the high measured mixing ratios of FA and AA.





First, we will consider the equilibrium partitioning of FA and AA between soil pore water and the atmosphere. The water-air partitioning of FA and AA can be estimated from Eq. (3) using measured soil temperature and pH, effective Henry's Law constants, and estimated soil formate and acetate concentrations (formate and acetate were chosen as 2 mg L$^{-1}$, probably an upper limit for this region (Nielsen et al., 2017; Ström et al., 2012)). Further uncertainty is introduced by using literature values

for soil formate and acetate concentrations because it requires making the assumption that all the formate and acetate extracted from the soils in those studies resided in the soil pore water, which is unlikely to be the case. The results of this estimate are shown in Table 4. The soil reservoirs of formate and acetate would need to be one or two orders of magnitude larger than they have been measured to be in similar environments in order to contribute more than a few percent to FA and AA levels. A very acidic soil would also promote the emission of FA and AA, but the soil samples were taken from three different micro-

environments (albeit all within ∼5 km of the station), and all samples had near-neutral pH. Additionally, other work examining Arctic soils has generally found near-neutral pH (Brummell et al., 2012; Nielsen et al., 2017). While a combination of lower pH and higher soil reservoirs of formate and acetate could perhaps lead to somewhat larger emissions than those estimated here, we conclude that equilibrium soil-air partitioning is unlikely to account for the high mixing ratios of FA and AA observed in this study.

During the time of highest FA and AA mixing ratios towards the beginning of the campaign (i.e., June 20 - June 26, Fig. 3), an excursion of FA and AA mixing ratios to extremely large values (11 and 40 ppbv respectively, Fig. 4) was observed. As this excursion occurred during the time of snow melt (Fig. 1b and c), we propose that it may have formed part of a spring emissions pulse such as has been observed for nitrous oxide (Wagner-Riddle et al., 2008). An emissions pulse associated with a spring thaw might result from increased microbial activity during soil thawing or from partitioning to the atmosphere of a pool

that has built up at depth and can be released once the snow has melted. Additionally, FA and AA emissions or mixing ratios have been observed previously to be enhanced during or after precipitation events, sometimes to quite large values similar to what we observed (Sanhueza and Andreae, 1991; Warneke et al., 1999; Yuan et al., 2015). One proposed explanation is that microbial production could be activated by the precipitation (Sanhueza and Andreae, 1991; Paulot et al., 2011). Pulses of FA or AA within the soil due to such a process would be transient, and might thus be quite different from the soil concentrations

measured in the studies cited here. A chemical explanation for emission pulses has also been proposed. The suggestion is that adsorbed polar molecules could be flushed out during an influx of water (such as during a precipitation event) as the *more* polar water molecules are preferentially taken up onto the adsorption sites (Warneke et al., 1999; Goss et al., 2004). Additionally, "pulsing" behaviour could arise from momentary pH changes in the soil during precipitation, allowing for the emission of FA and AA before the soil is buffered back to a more neutral pH. We suggest that one or more of these pulsing mechanisms might

provide an explanation for the unexpected increases of FA and AA during precipitation events, and potentially also for the very high mixing ratios observed on 20 June 2016. We wish to emphasize that further studies of direct emissions of FA and AA from soils, including the effects of precipitation, would be highly beneficial.



### 3.4.6 Plant emissions

To our knowledge, no information on the direct emissions of FA and AA from Arctic flora exists in the literature. Oxygenated VOCs (OVOCs), such as FA and AA, are produced as by-products of plant metabolism (Niinemets et al., 2014), and can either be emitted to the atmosphere or consumed by further metabolic processes. The direct emission of FA and AA from plants has

been documented (Kesselmeier et al., 1998; Kuhn et al., 2002; Guenther et al., 2000), but is not generally considered to be an important source of these compounds to the atmosphere globally (although emission estimates vary widely) (Paulot et al., 2011). A study of FA and AA in the central Amazon found bidirectional exchange of FA and AA between the forest canopy and the atmosphere (Jardine et al., 2011), with net upwards flux occurring in the absence of other sources, but net deposition when the atmosphere was impacted by advection of biomass burning plumes containing FA and AA. Hence whether the tundra

was acting as a source or a sink of FA and AA during our measurement campaign likely depends on what other sources were contributing to the ambient levels of those acids.

That the current understanding of the secondary photochemical production of FA and AA is not sufficient to explain observations has been thoroughly established (Paulot et al., 2011; Yuan et al., 2015; Millet et al., 2015; Liggio et al., 2017). Despite the understanding of these processes being incomplete, it is clear that the photo-oxidation of BVOCs such as isoprene

and monoterpenes is a very large source of FA and AA (Paulot et al., 2011, and references therein). The MEGANv2.1 model (Guenther et al., 2012), which is widely used to estimate BVOC emissions globally, predicts very low BVOC emissions in the subarctic and no emissions at all in large parts of the Arctic, which the model classifies as bare soil. There are two problems with this. First, the High Arctic is not only bare soil, with even polar desert areas reaching $30\,\%$ vegetation coverage (Liu and Treitz, 2016), and ecosystem-scale BVOC emissions comparable to those in the subarctic have been measured from High

Arctic vegetation (Schollert et al., 2013; Rinnan et al., 2014; Vedel-Petersen et al., 2015; Lindwall et al., 2016). Second, soil emissions of reactive alkanes have been inferred from ecosystem emission measurements (Schollert et al., 2013), raising the possibility that emissions of BVOCs from bare soil should also be considered in models. Because BVOC emissions are often temperature-dependent (Guenther et al., 2012), it is also worth noting that soil temperatures at $10\,\mathrm{cm}$ depth at times exceeded air temperatures (Fig. 3). The temperature a few centimeters above the ground in the summer Arctic can reach up to $10\,^{\circ}\mathrm{C}$

higher than routinely measured air temperatures (Schollert et al., 2013), perhaps explaining why emission measurements in the Arctic seem to be higher than those estimated from temperature-dependent parameterizations.

The emission of many plant volatiles is controlled to some extent by light and temperature (Guenther, 2013), and thus the mixing ratios of BVOCs commonly display distinct diurnal profiles. Hence the diurnal variability in FA and AA might point to a primary or secondary plant source for these acids, although the suggestion of an upward gradient in FA and AA (Section

3.4.1) makes it less likely that a secondary source is responsible for these observations. An upward gradient might be less likely to develop from a secondary source, as FA and AA production from precursors takes several hours at least (Liggio et al., 2017), allowing time for the precursor BVOCs to be mixed upward into the boundary layer. As we did not measure BVOC mixing ratios during this campaign, we cannot estimate the contribution of their oxidation to the FA and AA we measured. However, given that Arctic plants are known to emit BVOCs, it would be surprising if photo-oxidation of BVOCs did *not* contribute to





some extent to FA and AA mixing ratios in the summer boundary layer. Measurements of precursors in conjunction with FA and AA would be useful to further explore this question.

## 4    Conclusions

We measured gas-phase mixing ratios of formic and acetic acids as well as formate and acetate concentrations in precipitation
over a three-week period during summer 2016 at Alert, Nunavut. We observed high, and highly variable, mixing ratios of FA and AA, which we interpret as arising from regional sources within the Arctic. In particular, the surprisingly large magnitude of the mixing ratios is suggestive of not just a shallow boundary layer but of strong vertical gradients resulting from a surface source. The first half of the campaign was relatively cold, wet, and overcast, and yet high mixing ratios of FA and AA were observed, with increases during and immediately following precipitation events. These observations indicate that high levels
of FA and AA exist in a moist environment where the pH of soil and precipitation are larger than the pKa of these acids, that is, an environment in which physical equilibrium partitioning should not favor re-volatilization of FA and AA. Pulses of FA and AA from wet soil have been observed previously (Warneke et al., 1999; Sanhueza and Andreae, 1991), suggesting that similar biological or chemical mechanisms are at play in the Arctic environment. The second half of the campaign was relatively warm and sunny, and the FA and AA mixing ratios displayed diurnal cycles, suggesting the influence of photo-oxidation, whether of
plant-emitted BVOCs or heterogeneous oxidation of surfaces. The relative importance of these various emission mechanisms likely changed over the study period in concert with the dramatic changes in the environmental conditions characteristic of the short Arctic summer.

Overall, this study supports the growing understanding that formic acid is ubiquitously produced and displays high mixing ratios in most environments where it is measured, and suggests the same for acetic acid. Furthermore, the observation of high
mixing ratios under very different environmental conditions – cold, wet, and overcast versus warm and sunny – highlight that FA and AA emission processes appear to be varied and complex. Furthermore, the most important processes appeared to be changing over the course of the campaign. These processes and their magnitudes are not adequately represented in chemical transport models, as evidenced by large underestimates of FA and AA mixing ratios in the region (Paulot et al., 2011). To work towards ameliorating the models, the next step should be to repeat these observations over a longer time period, both
before the melt period and after, to better allow elucidation of the changing sources. For measurements performed at Alert, efforts to diagnose the relative impacts of different meteorological processes (i.e., turbulent mixing versus advection) on the observed mixing ratios would be valuable. The measurement of vertical fluxes is also highly desirable, particularly with respect to inclusion in models. In conclusion, while it is clear that models are not capturing the processes affecting FA and AA in the Arctic, more information is required before those deficiencies can be addressed.

*Data availability.*    NETCARE (Network on Climate and Aerosols: Addressing Key Uncertainties in Remote Canadian Environments, http://www.netcare-project.ca), which organized the field campaigns described in this work, is moving toward a publically available, online data archive. In



the meantime, data can be accessed by contacting the principal investigator of the network: Jonathan Abbatt, University of Toronto (jabbatt@chem.utoronto.ca).

*Author contributions.* J. Liggio, J.P.D. Abbatt, and J. Murphy designed the experiment. E.L. Mungall and J.J.B. Wentzell collected the gas-phase data. G. Wentworth collected the precipitation and soil data. S. Sharma facilitated the measurements. E.L. Mungall performed the
5    calibrations and analyzed the data. D. Kunkel ran the FLEXPART-ECMWF model. E. Gute provided meteorological expertise. D. Tarasick analyzed the radiosonde data. C.J. Cox collected and analyzed the pyranometer data. E.L. Mungall wrote the manuscript with contributions from all co-authors.

*Competing interests.* No competing interests are present.

*Acknowledgements.* The authors would like to acknowledge the financial support of NSERC for the NETCARE project funded under the
10    Climate Change and Atmospheric Research program. We gratefully acknowledge Richard Leaitch for facilitating the Alert campaign, as well as the GAW lab operator Kevin Rawlings and the co-op student Dana Stephenson. We are also grateful to everyone at CFS Alert, both military and Nasittuq, for their hospitality and help.



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
