# Peer review of "High gas-phase mixing ratios of formic and acetic acid in the High Arctic"

_Atmospheric Chemistry and Physics, 2018_

## Referee Comment (RC1) · Anonymous Referee #1 · 15 Mar 2018

Review of Mungall et al., "High gas-phase mixing ratios of formic and acetic acid in the High Arctic"

Mungall et al describe new measurements of the concentration of formic and acetic acid in the arctic. These are exceedingly rare measurements and a welcome addition to the literature. The concentrations of FA and AA are much larger than expected and the sources that sustain the elevated mixing ratios are at this point not abundantly clear. However, this work highlights that future studies in these regions need to be carried forward to better constrain the sources of acidic gases in the high Arctic. The paper should be published following the authors attention to the following brief comments.

General Comments

I was surprised there was not a stronger focus on the use of the formic to acetic acid ratio in this study, given that the loss rates for FA and AA are similar and there are existing measurements of this ratio that can be compared to.

Specific and technical comments:

Page 2, Line 23: Does the lifetime estimate of 1-2 days pertain to the arctic conditions discussed here? If loss is driven by deposition, I would have expected more variance in this estimate based on season/location. It would be helpful for the authors to discuss this in more detail.

Page 3, Line 10: It is not clear how the absolute humidity was measured in the flow that exits the IMR based on the standard configuration of a CI-ToFMS. Was this measured in the bypass line?

Page 6, line 6: How often were radiosondes deployed?

Figure 3 and 5: Why is there more high frequency variance in the AA measurement than the FA measurement? Is the S/N ratio for FA higher than that for AA at the same atmospheric concentration, due to differences in sensitivity? Is atmospheric FA more variable at high frequency than AA? A short comment on the source of this apparent difference would be helpful.

Page 9, line 15: Can estimates of the mixing heights for Alert during this period be given?

Page 11, Line 15: The lack of suppression in FA or AA during the extended precipitation events is surprising/confusing. I can appreciate that a source could remain during the beginning of the rain storm, but it is not clear how this can persist as surface soil water content increases (I assume the ground is wet). Are there hints from other acids measured by the CIMS that this is specific to FA/AA? Presumably HNO3 should be completely titrated for the longer events. Finally, can the references be separated for those that have seen enhancements during precipitation as opposed to those following

precipitation as these are two very different ideas/processes.

---

## Referee Comment (RC2) · Anonymous Referee #2 · 6 Apr 2018

This is a well-written and thoughtful manuscript that describes high FA and AA in the Arctic. The authors note some interesting features in the data (correlation to high wind speeds), and then move to examining the sources. Anthropogenic activity is not a source (while the authors have few tracers for anthropogenic activity, I am satisfied by the use of aerosol number in this environment). The authors further demonstrate that partitioning from the snowpack is unlikely using back-of-the-envelope calculations. Attribution of flux to soil emissions is more challenging due to the lack of data, but the authors present a balanced picture of the relevant literature. Similarly, plants are an unlikely source of formic acid. Overall, the authors attribute the elevated formic and acetic acid concentrations to a combination of sources, but cannot point to a clear source or specific insight. However, I suspect this gets to the core of the challenge of modeling

organic acids: there is no single missing source or single problem: instead, multiple equilibria control the system. Thus, I think this paper provides new information and is presented in a robust manner. It warrants publication in ACP with minor corrections.

Specific comments:

Regarding iodide ionization: - P2, l22: Ionization is generally thought to be a ligand-exchange reaction, not a clustering reaction. The end product is a cluster (or adduct). The authors allude to this later in the section, but as written, I don't quite agree with this sentence.

- P4, l8: my understanding is that water vapor can enhance the ionization efficiency for some molecules and suppress it for others – particularly at low RH

Figure 4. I don't understand where these data come from. Could you be more specific about what you mean by 'greatest FA and AA mixing ratios', and exactly what data went into this figure. Arbitrarily choosing to only show a diurnal profile for high mixing ratio days seems quite selective, and I instead recommend a more robust line of reasoning – for example 'wind speed > x' or somesuch.

Technical corrections

P2 l28: should read 'source that exceeds'

Figure captions are inconsistent on the use of 'formic acid' vs 'FA' and 'acetic acid' vs 'AA'. Please be consistent.

Dates are in UTC in the first figures, and then local time in Figure 6 (though 6a denotes local time and 6c does not). This is quite confusing to the reader, and it would be helpful to either include both times, a clear conversion, or – preferably – all the data in local time. The problem with UTC is that I have no sense of what is happening to solar radiation or when daylight is. If the authors wish to continue with UTC, a clear conversion for UTC to local time AND a timetrace of solar radiation or actinic flux would be useful. Overall, though, the authors should be consistent.

P19, l7: should read "of not only a shallow boundary layer, but also of "

---

## Referee Comment (RC3) · Anonymous Referee #3 · 29 Apr 2018

Mungall et al. present field observations of formic and acetic acid in the gas-phase, and of formate and acetate in rain water samples taken at Alert, Nunavut during a the early summer 2016. Surprisingly high and variable gas-phase concentrations are at odds with our current understanding about sources of biogenic VOC precursor molecules, or possible other sources in the remote arctic atmosphere.

The manuscript is generally well written, and the topic falls well within the scope of ACP. The work is of considerable quality, and leverages state-of-the-art instrumentation. I recommend publication after the below points have been addressed.

Comments: 1) The abstract points out that the scarcity of measurements at high latitudes is a hindrance to understanding these acids and their sources. Yet the authors seem to be making a conscious choice to limit discussion to previous measurements

[Figure]

of FA and AA. Why? In terms of the sources for acids, it is not obvious why there should be anything special about FA and AA, other than that they reside primarily in the gas-phase. There is a rich history of acid measurements in the condensed phase at Alert, that dates prior to the advent of online techniques, and includes measurements of annual cycles of organic acids in aerosols in 1987-1988 (published approximately a decade later). Early work by Kimi Kawamura in aerosols, and also Len Barrie comes to mind. A thorough literature review might reveal other information. This should be discussed. A paragraph that summarizes what other acids measurements have revealed about sources of other acids at Alert is currently missing.

2) The lifetime of FA and AA is listed as 1-2 days in the introduction. Does this apply to the conditions at Alert? How different are the environmental conditions characterized in Paulot et al 2010 to those during the wet/cool period, and to the photochemical period? Variables such as boundary layer height, dry and wet deposition rates are likely quite different for the different periods studies, and a shallow boundary layer at Alert may affect that lifetime?

3) The experimental section mentions estimates of mixing height from radiosondes deployed by the EC weather station at Alert. At what frequency are these observations available? What is known about the boundary layer height during the cold/wet and photochemical period? What gaps are there in our understanding of boundary layer height at Alert? A thorough discussion of boundary layer height is missing, and is not only relevant to lifetimes (see point 2); also the dilution of a surface source flux, the observed wind-speed dependence of the photochemical cycle, previous observations of extremely shallow inversions (few m-10m) above ground in the arctic point to a key role of boundary layer height, etc. Please expand this discussion. Data of mixing height is missing in Figure 3.

4) Figure 4: Maybe I missed it, but was the acid spike in Figure 4 observed with high or low light conditions? Maybe this could be clarified by using a second y-axis to illustrate a subset of other variables. It is later suggested (page 17, line 29) that changes in pH

due to precipitation may drive atmospheric variability of acid concentrations here. A reference to Figure 4 seems appropriate here (currently missing) - and revising Figure 4 would benefit from showing other parameters to support the claim, if at all possible.

5) A minor point, but somewhat disconcerting: Table 3 lists R2 for FA*WS as 0.8, while Figure 6d lists that same number, for the same period as R2 = 0.87 - which is correct?

6) Is there no data about the enrichment factor of acids at surfaces of snow crystals, or as a result of freeze thaw cycles? Concentration is likely, and well established for halogens (e.g. frost flower mechanism of halogen activation). The assumptions underlying estimates of gas-phase acid concentrations on page 15, lines 28f seem very crude. How much acid can realistically be expected to be rationalized if the assumption of equilibrium partitioning is refined, or conversely, how much of a change in pH, or surface concentration enrichment is needed to explain the observations?

7) There is no mentioning of aerosols as a source for FA and AA. Why is this? Can the authors rule out that aerosols are incubators for FA and AA formation?

---

## Author Comment (AC1) · 27 Jun 2018

We thank the reviewers for their insight and helpful comments and corrections.

All pages and line numbers correspond to the marked-up version of the manuscript (i.e., manuscript with tracked changes shown).

**1   Reviewer 1**

**RC1:** *"I was surprised there was not a stronger focus on the use of the formic to acetic acid ratio in this study, given that the loss rates for FA and AA are similar and there are*

[Figure]

*existing measurements of this ratio that can be compared to."*

**AC1:** This is a good point, and is definitely something we investigated during our analysis. However, we did not find any robust association between changes in the ratio (which are not dramatic) and other variables. Furthermore, as we mentioned in the manuscript (page 9, lines 10–12), existing measurements of this ratio are inconclusive; i.e., there is no consensus in the literature as to what a higher or lower ratio indicates about the sources of these molecules. For this reason, we chose not to discuss the changes in the ratio.

Text concerning the ratio in the manuscript (page 9, lines 9–12): "The decrease in AA across the campaign may indicate a shift to sources with a higher FA to AA ratio. There are some indications that a higher FA to AA ratio is associated with biogenic sources of these acids (Khare et al., 1999; Talbot et al., 1988), but as no conclusive evidence exists we have not attempted to use this changing ratio to further our analysis."

**RC2:** *"Page 2, Line 23: Does the lifetime estimate of 1-2 days pertain to the arctic conditions discussed here? If loss is driven by deposition, I would have expected more variance in this estimate based on season/location. It would be helpful for the authors to discuss this in more detail."*

**AC2:** Thanks for pointing out this omission. It is indeed likely that the lifetimes of FA and AA are quite short in the summer Arctic, owing to the shallow boundary layer and frequency of wet deposition. We have added further discussion of lifetime to the text and acknowledged that it is likely to be quite short.

**Manuscript Changes:**

Page 2, lines 23–25: "Considering the various sinks, the average global lifetimes of FA and AA in the boundary layer have been estimated at 1–2 days (Paulot et al., 2011), although the lifetimes are of course highly dependent on the likelihood of deposition

and thus, for example, on how dry or wet the environment is."

Page 13, lines 7–11: "The lifetimes of FA and AA are also likely to have changed over time. In particular, their lifetimes were longer once the ground dried up post-snow-melt, reducing the impact of deposition. However, in the case of AA, this increase in lifetime may have been at least partially counteracted by the effect of increased sunlight during the latter, brighter half of the campaign, which may have led to increases in photo-oxidation and a corresponding increase in the lifetime of AA."

**RC3:** *"Page 3, Line 10: It is not clear how the absolute humidity was measured in the flow that exits the IMR based on the standard configuration of a CI-ToFMS. Was this measured in the bypass line?"*

**AC3:** Thank you for pointing out this omission. We have added a few sentences explaining how the CIMS was configured.

**Manuscript Changes:**

Page 4, lines 7–9: "The pressure in the IMR was controlled via a flow-controller mediated leak in the line leading from the IMR to the pump. A relative humidity and temperature sensor was installed in this line to monitor the absolute humidity in the IMR."

**RC4:** *"Page 6, line 6: How often were radiosondes deployed?"*

**AC4:** Thanks for pointing out this omission. The radiosondes were deployed twice daily. We have added this detail to the method section and added further detail concerning the radiosondes and the mixing height determination to the supporting information.

**Manuscript Changes:**

Page 6, lines 14–16: "Mixing height was estimated from radiosoundings deployed by the EC weather station at Alert twice-daily, at 0 and 12 UTC (0700 and 1900 local time). Further details on mixing height estimation may be found in Supporting Information Section S2."

**RC5:** *"Figure 3 and 5: Why is there more high frequency variance in the AA measurement than the FA measurement? Is the S/N ratio for FA higher than that for AA at the same atmospheric concentration, due to differences in sensitivity? Is atmospheric FA more variable at high frequency than AA? A short comment on the source of this apparent difference would be helpful."*

**AC5:** Thanks for this comment. Indeed, the S/N ratio is much higher for FA than for AA, due to differences in sensitivity. We now address this difference in the methods section.

**Manuscript Changes:**

Page 5, lines 10–12: "Due to the instrumental response (i.e., sensitivity) being much higher for FA than for AA, the FA signal had a much higher signal-to-noise ratio than did the AA signal, resulting in the high-frequency variance visible in the AA time series."

**RC6:** *"Page 9, line 15: Can estimates of the mixing heights for Alert during this period be given?"*

**AC6:** Thanks for pointing out this oversight. We have revised the manuscript to include considerably more discussion of boundary layer height, hopefully clarifying this portion of the text.

**Manuscript Changes:**

Page 10, lines 6–26: "Determination of the boundary-layer mixing height at Arctic locations is made complex by the interactions of snow-covered surfaces, sea ice and open

water (Anderson and Neff, 2008). While the unstable conditions of well-mixed surface boundary layers do occur in summer, they are less common at Alert than statically stable conditions, where the potential temperature increases monotonically with height, so that mixing is suppressed. Typically the rate of increase of potential temperature is large enough that the physical temperature increases, leading to a surface-based temperature inversion (SBI). SBIs are generally found even when an unstable mixing layer exists at a lower altitude. SBIs in polar regions have been the subject of a number of studies (Bourne et al., 2010; Aliabadi et al., 2016), and are often taken as an indicator of boundary layer height (Bradley et al., 1993; Seidel et al., 2010). Although it is not obvious why the depth of an SBI should correspond to the depth of a mixed surface layer, as the strong gradient of potential temperature should strongly suppress mixing, rather than encourage it, ozone soundings at Alert and other Arctic sites typically show strong gradients of ozone and relative humidity at this height (Tarasick and Bottenheim, 2002). However, this may reflect previous boundary layer mixing and subsequent transport, possibly from the nearby ice zone, or differential transport of layers (Anderson and Neff, 2008).

Since the sources of formic and acetic acid are land-based and presumed local (unlike surface ozone depletion events) the relevant parameters are the convective mixing height, where it exists, and the vertical gradient of potential temperature, as a measure of the resistance to mixing (rather than the SBI height, which may represent non-local conditions). These were calculated from twice-daily radiosoundings at Alert (Figure S5). In general, more stable conditions exist during the low wind speed regime. A convective mixing height is found for only 60% of all soundings, but where it exists, it averages 189 m, compared to 440 m during the high wind speed regime. A similar difference is evident in the gradient of potential temperature: between the surface and 1 km, the change in potential temperature is 7.5 K during the low wind speed regime, and 4.0 K during the high wind speed regime. In either case the difference is approximately a factor of 2. SBI heights are also greater during the high wind speed regime (803 vs 595 m)."

**RC7:** *"Page 11, Line 15: The lack of suppression in FA or AA during the extended precipitation events is surprising/confusing. I can appreciate that a source could remain during the beginning of the rain storm, but it is not clear how this can persist as surface soil water content increases (I assume the ground is wet). Are there hints from other acids measured by the CIMS that this is specific to FA/AA? Presumably HNO3 should be completely titrated for the longer events. Finally, can the references be separated for those that have seen enhancements during precipitation as opposed to those following precipitation as these are two very different ideas/processes."*

**AC7:** We completely agree that the lack of suppression is confusing. This section attempts to rationalize this confusing observation. We note, also, that the ground is very wet before, during, and after the precipitation events during the beginning of the campaign, due to snow melt. It is indeed difficult to reconcile such large (apparent) FA and AA mixing ratios with the very wet ground. This is what we try to do in this manuscript.

Unfortunately, despite the ability of the CIMS to measure a wide variety of acids, we were not able to quantify any acids other than FA and AA due to calibration difficulties. We agree that such information would be valuable to better understand the observations, but we just don't have it.

Thank you for the suggestion to separate the references; you are entirely correct that it should have been done that way initially. We have done so in the revised manuscript.

**Manuscript Changes:**

Page 12, lines 11–14: "Large enhancements of FA and AA mixing ratios during (Warneke et al., 1999; Yuan et al., 2015; Greenberg et al., 2012) or immediately following (Sanhueza and Andreae, 1991; Talbot et al., 1988; Greenberg et al., 2012) precipitation events have been reported previously."

**2 Reviewer 2**

**RC1:** *"Regarding iodide ionization: - P2, l22: Ionization is generally thought to be a ligandexchange reaction, not a clustering reaction. The end product is a cluster (or adduct). The authors allude to this later in the section, but as written, I don't quite agree with this sentence."*

**AC1:** Thanks for pointing this out. We have changed the wording in accordance with this suggestion.

**Manuscript Changes:**

Page 3, lines 23–25: "The iodide CIMS ionizes analyte molecules through a ligand-exchange reaction with the $I_2^- \cdot H_2O$ ion, leading to the formation of a charged adduct that is detected by the mass spectrometer (Lee et al., 2014)."

**RC2:** *"P4, l8: my understanding is that water vapor can enhance the ionization efficiency for some molecules and suppress it for others – particularly at low RH"*

**AC2:** Thanks, this is a good point. We have corrected the wording in the manuscript.

**Manuscript Changes:** Page 5, lines 2–5: "The humidity dependence of the individual compound sensitivities was determined, as water vapor can cluster preferentially with the iodide ions, leading to either an increase or a decrease in the ionization efficiency of analyte molecules depending on the identity of the molecule."

**RC3:** *"Figure 4. I don't understand where these data come from. Could you be more specific about what you mean by 'greatest FA and AA mixing ratios', and exactly what data went into this figure. Arbitrarily choosing to only show a diurnal profile for high mixing ratio days seems quite selective, and I instead recommend a more robust line of reasoning – for example 'wind speed > x' or somesuch."*

**AC3:** We apologize for the unclear wording in the caption. This is not a diurnal profile, but rather an expanded view of a particularly large excursion in FA and AA mixing ratios that occurred over a short period of time. We have modified the figure caption to make this clearer to the reader.

**Manuscript Changes:**

Page 9 (Figure 4): "Expanded view of the time series on June 20 2016, during which a very large excursion in FA and AA mixing ratios was observed. Note the factor of ten difference in the y-axis scale. This large excursion to high values took place during a light precipitation event under stagnant, ($\sim$1.5 ms$^{-1}$), cool ($\sim$4C, low-light (overcast) conditions."

**RC4:** *"P2 l28: should read 'source that exceeds'"*

**AC4:** Thank you for this correction. The change has been made.

**Manuscript Changes:** Page 2, line 29: "source that exceeds"

**RC5:** *"Figure captions are inconsistent on the use of 'formic acid' vs 'FA' and 'acetic acid' vs 'AA'. Please be consistent."*

**AC5:** Thanks for catching this inconsistency. We have made the necessary changes.

**Manuscript Changes:** Page 9 (Figure 4): "a very large excursion in FA and AA mixing ratios"

**RC6:** *"Dates are in UTC in the first figures, and then local time in Figure 6 (though 6a denotes local time and 6c does not). This is quite confusing to the reader, and it would be helpful to either include both times, a clear conversion, or – preferably – all the data in local time. The problem with UTC is that I have no sense of what is happening to*

*solar radiation or when daylight is. If the authors wish to continue with UTC, a clear conversion for UTC to local time AND a timetrace of solar radiation or actinic flux would be useful. Overall, though, the authors should be consistent."*

**AC6:** We apologize for the confusion. The rationale behind using local time in Figure 6 is that this figure is specifically related to diurnal variability, and thus we found that the figures could be interpreted more intuitively when the x-axis was in local time. That being said, local time is a bit of a tricky concept in the High Arctic, making this a somewhat arbitrary choice. While we do wish to keep Figure 6 in local time to improve readability of the figure, we have updated the figure to reflect the conversion to UTC to improve consistency. UTC is used elsewhere in the manuscript because it is the standard. We note that Figure 3 does include a solar radiation time series, and that as Figure 4 is a subset of Figure 3, the information can be easily found.

**Manuscript Changes:** Updated x-axes in Figure 6.

**RC7:** *"P19, l7: should read 'of not only a shallow boundary layer, but also of'"*

**AC7:** Thanks for the correction. We have made the necessary change.

**Manuscript Changes:** Page 20, line 11: "of not only a shallow boundary layer, but also of"

**3   Reviewer 3**

**RC1:** *"The abstract points out that the scarcity of measurements at high latitudes is a hindrance to understanding these acids and their sources. Yet the authors seem to be making a conscious choice to limit discussion to previous measurements of FA and AA. Why? In terms of the sources for acids, it is not obvious why there should be anything*

*special about FA and AA, other than that they reside primarily in the gas-phase. There is a rich history of acid measurements in the condensed phase at Alert, that dates prior to the advent of online techniques, and includes measurements of annual cycles of organic acids in aerosols in 1987-1988 (published approximately a decade later). Early work by Kimi Kawamura in aerosols, and also Len Barrie comes to mind. A thorough literature review might reveal other information. This should be discussed. A paragraph that summarizes what other acids measurements have revealed about sources of other acids at Alert is currently missing."*

**AC1:** We chose to limit our discussion to FA and AA in part because we were not successful in making measurements of any other acids, in part because we do think that these acids are unusual, and in part because we did not feel the previous acid measurements to be relevant to this study.

First, FA and AA are unusual in their wide diversity of sources, as we discuss in the manuscript. Most larger acids have a more restricted range of sources. Relatedly, the fact that FA and AA do primarily reside in the gas phase means that their transport patterns may be quite different from aerosol-bound species. It would have been very interesting to see how FA and AA related to other acids, and such information might have revealed more information about the sources of these molecules than we were able to glean in the present study. However, we simply do not have those measurements.

Second, the previous work at Alert was almost entirely focused on the spring (i.e., the polar sunrise) and was done in the aerosol phase. There is so little aerosol mass in summer that aerosols are a very unlikely source for the large mixing ratios of FA and AA we observed during this campaign.

**RC2:** *"The lifetime of FA and AA is listed as 1-2 days in the introduction. Does this apply to the conditions at Alert? How different are the environmental conditions characterized*

*in Paulot et al 2010 to those during the wet/cool period, and to the photochemical period? Variables such as boundary layer height, dry and wet deposition rates are likely quite different for the different periods studies, and a shallow boundary layer at Alert may affect that lifetime?"*

**AC2:** Thanks for making these points. The lifetime quoted in the introduction from Paulot et al. 2010 is a global average. We have made this clearer in the introduction. We have also added a bit more discussion, as you suggest, of how the environmental changes across the campaign may have affected the sinks of FA and AA. Overall, the specific lifetime of these compounds is not crucial to our discussion: all that matters is that the lifetimes are sufficiently short that only local sources need be discussed. As the factors specific to Alert (e.g., shallow mixing height) are not likely to prolong the lifetimes, the lack of a sophisticated understanding of lifetime is perhaps not the major uncertainty in the discussion presented here.

**Manuscript Changes:**

Page 2, lines 23–25: "Considering the various sinks, the average global lifetimes of FA and AA in the boundary layer have been estimated at 1–2 days (Paulot et al., 2011), although the lifetimes are of course highly dependent on the likelihood of deposition and thus, for example, on how dry or wet the environment is."

Page 13, lines 7–11: "The lifetimes of FA and AA are also likely to have changed over time. In particular, their lifetimes were longer once the ground dried up post-snow-melt, reducing the impact of deposition. However, in the case of AA, this increase in lifetime may have been at least partially counteracted by the effect of increased sunlight during the latter, brighter half of the campaign, which may have led to increases in photo-oxidation and a corresponding increase in the lifetime of AA."

**RC3:** *"The experimental section mentions estimates of mixing height from radiosondes deployed by the EC weather station at Alert. At what frequency are these observations*

[Figure]

*available? What is known about the boundary layer height during the cold/wet and photochemical period? What gaps are there in our understanding of boundary layer height at Alert? A thorough discussion of boundary layer height is missing, and is not only relevant to lifetimes (see point 2); also the dilution of a surface source flux, the observed wind-speed dependence of the photochemical cycle, previous observations of extremely shallow inversions (few m-10m) above ground in the arctic point to a key role of boundary layer height, etc. Please expand this discussion. Data of mixing height is missing in Figure 3."*

**AC3:** Thanks for these comments. The radiosondes were deployed twice daily. We have added this detail to the method section (thanks for pointing out the omission) and have added further detail concerning the mixing height estimates to the Supporting Information. We agree entirely that boundary layer height is a crucial variable, and have expanded our discussion in the manuscript. Unfortunately, we cannot say as much about boundary layer height as we would like, given the complexity of determining it in this location and the limited observations available. For this reason, we restrict our analysis to the observation that mixing heights tended to be higher during the high wind speed regime, which may explain the lower mixing ratios observed on average during this time. We do not feel that the data support a more in-depth analysis, as useful as that would be. This is also why we do not include the mixing height data in Figure 3: we feel that given the complexity of interpreting these numbers, and how sporadic the observations are, they would simply clutter the figure and encourage over-interpretation of the results.

Finally, as to the extremely shallow inversions, our understanding is that such low inversion heights are prevalent during polar night, not under summer, sunlit conditions.

**Manuscript Changes:**

Page 6, lines 14–16: "Mixing height was estimated from radiosoundings deployed by the EC weather station at Alert twice-daily, at 0 and 12 UTC (0700 and 1900 local time).

Further details on mixing height estimation may be found in Supporting Information Section S2."

Page 10, lines 6–26: "Determination of the boundary-layer mixing height at Arctic locations is made complex by the interactions of snow-covered surfaces, sea ice and open water (Anderson and Neff, 2008). While the unstable conditions of well-mixed surface boundary layers do occur in summer, they are less common at Alert than statically stable conditions, where the potential temperature increases monotonically with height, so that mixing is suppressed. Typically the rate of increase of potential temperature is large enough that the physical temperature increases, leading to a surface-based temperature inversion (SBI). SBIs are generally found even when an unstable mixing layer exists at a lower altitude. SBIs in polar regions have been the subject of a number of studies (Bourne et al., 2010; Aliabadi et al., 2016), and are often taken as an indicator of boundary layer height (Bradley et al., 1993; Seidel et al., 2010). Although it is not obvious why the depth of an SBI should correspond to the depth of a mixed surface layer, as the strong gradient of potential temperature should strongly suppress mixing, rather than encourage it, ozone soundings at Alert and other Arctic sites typically show strong gradients of ozone and relative humidity at this height (Tarasick and Bottenheim, 2002). However, this may reflect previous boundary layer mixing and subsequent transport, possibly from the nearby ice zone, or differential transport of layers (Anderson and Neff, 2008).

Since the sources of formic and acetic acid are land-based and presumed local (unlike surface ozone depletion events) the relevant parameters are the convective mixing height, where it exists, and the vertical gradient of potential temperature, as a measure of the resistance to mixing (rather than the SBI height, which may represent non-local conditions). These were calculated from twice-daily radiosoundings at Alert (Figure S5). In general, more stable conditions exist during the low wind speed regime. A convective mixing height is found for only 60% of all soundings, but where it exists, it averages 189 m, compared to 440 m during the high wind speed regime. A similar

difference is evident in the gradient of potential temperature: between the surface and 1 km, the change in potential temperature is 7.5 K during the low wind speed regime, and 4.0 K during the high wind speed regime. In either case the difference is approximately a factor of 2. SBI heights are also greater during the high wind speed regime (803 vs 595 m)."

**RC4:** *"Figure 4: Maybe I missed it, but was the acid spike in Figure 4 observed with high or low light conditions? Maybe this could be clarified by using a second y-axis to illustrate a subset of other variables. It is later suggested (page 17, line 29) that changes in pH due to precipitation may drive atmospheric variability of acid concentrations here. A reference to Figure 4 seems appropriate here (currently missing) - and revising Figure 4 would benefit from showing other parameters to support the claim, if at all possible."*

**AC4:** Thanks for these comments. The reason we don't have other variables in Figure 4 is that as it covers such a short time period, the other variables didn't display much variability, so adding them to the figure essentially adds a bunch of flat lines. The point that it would be helpful to have a sense of these variables is well taken, though, so we have added some general points to the figure caption concerning the atmospheric conditions at that time. We have also added the reference to Figure 4 where you suggested—thank you for catching this omission.

**Manuscript Changes:**

Page 9 (Figure 4): "Expanded view of the time series on June 20 2016, during which a very large excursion in FA and AA mixing ratios was observed. Note the factor of ten difference in the y-axis scale. This large excursion to high values took place during a light precipitation event under stagnant, ($\sim$1.5 ms$^{-1}$), cool ($\sim$4C, low-light (overcast) conditions."

Page 19, line 3: "... the very high mixing ratios observed on June 20 2016 (Fig. 4)."

**RC5:** *"A minor point, but somewhat disconcerting: Table 3 lists R2 for FA\*WS as 0.8, while Figure 6d lists that same number, for the same period as R2 = 0.87 - which is correct?"*

**AC5:** The $R^2$ values in Table 3 and Figure 6 are for different equations. Table 3 presents the results from the multiple linear regression of FA on wind speed and solar radiation, while in Figure 6 the product of FA and wind speed is regressed on solar radiation. These are different equations and as they are equivalent, there is no reason for their $R^2$ values to match. Table 3 tells us that wind speed and solar radiation together explain 80% of the variability in FA, while Figure 6d tells us that solar radiation can explain 87% of the variability in the product of FA and wind speed.

While I agree that it may be confusing to have these different but related quantities in the manuscript, they are there for different reasons. The multiple linear regression is easy to interpret mathematically, while it is not clear what exactly the product of FA and wind speed signifies. However, I include the latter because it allows us to visualize the relationship between these three quantities. Intuitively, the modulation of the FA mixing ratio by wind speed makes sense.

**RC6:** *"Is there no data about the enrichment factor of acids at surfaces of snow crystals, or as a result of freeze thaw cycles? Concentration is likely, and well established for halogens (e.g. frost flower mechanism of halogen activation). The assumptions underlying estimates of gas-phase acid concentrations on page 15, lines 28f seem very crude. How much acid can realistically be expected to be rationalized if the assumption of equilibrium partitioning is refined, or conversely, how much of a change in pH, or surface concentration enrichment is needed to explain the observations?"*

**AC6:** While we were not able to find any information specific to FA and AA about enrichment at snow surfaces, we refer the reviewer to the studies referred to in the manuscript (Kuhn, 2001; Meyer et al., 2009; page 17, line 11), which present work that

has been done on other negative ions in these contexts.

The assumptions underlying the estimates of gas-phase acid concentrations are indeed crude. However, as we had no other information to work with, we felt that more detailed assumptions would imply an inappropriate level of certainty. For this reason, we did not put numbers to the question of how, precisely, enrichment or pH changes would affect these estimates. Instead, we explain that "a three order of magnitude increase in concentration as a result of freeze-thaw cycles or solute exclusion would be required for equilibrium partitioning to explain our observations." (page 17, lines 14–15) As pH changes would be effected by increases in concentration of acidic ions such as $SO_4^{2-}$, this statement applies both to increases in concentration of FA and AA and to decreases in pH.

**RC7:** *"There is no mentioning of aerosols as a source for FA and AA. Why is this? Can the authors rule out that aerosols are incubators for FA and AA formation?"*

**AC7:** FA and AA concentrations in the particle phase are generally only a few percent of gas-phase mixing ratios (Liu et al., 2012; Chebbi and Carlier, 1996; Baboukas et al., 2000; page 2, lines 20–22). Given the very high gas-phase mixing ratios we measured at Alert and the very low particle loadings characteristic of the summer Arctic, we believe that the contribution from the particle phase was likely insignificant during the campaign.

**Manuscript Changes:**

[revised manuscript text omitted]